# Real-Time Fluorescence Microscopy on Living *E. coli* Sheds New Light on the Antibacterial Effects of the King Penguin β-Defensin AvBD103b

**DOI:** 10.3390/ijms23042057

**Published:** 2022-02-12

**Authors:** Céline Landon, Yanyu Zhu, Mainak Mustafi, Jean-Baptiste Madinier, Dominique Lelièvre, Vincent Aucagne, Agnes F. Delmas, James C. Weisshaar

**Affiliations:** 1Department of Chemistry, University of Wisconsin-Madison, Madison, WI 53706, USA; zhuyanyupku@gmail.com (Y.Z.); mainak.mustafi@gmail.com (M.M.); weisshaar@chem.wisc.edu (J.C.W.); 2Center for Molecular Biophysics, CNRS, 45071 Orléans, France; jean-baptiste.madinier@cnrs-orleans.fr (J.-B.M.); dominique.lelievre@cnrs-orleans.fr (D.L.); vincent.aucagne@cnrs-orleans.fr (V.A.); delmas@cnrs-orleans.fr (A.F.D.)

**Keywords:** disulfide-rich peptide, antimicrobial peptide, defensin, all—D peptide, fluorescence microscopy, live bacteria

## Abstract

(1) Antimicrobial peptides (AMPs) are a promising alternative to conventional antibiotics. Among AMPs, the disulfide-rich β-defensin AvBD103b, whose antibacterial activities are not inhibited by salts contrary to most other β-defensins, is particularly appealing. Information about the mechanisms of action is mandatory for the development and approval of new drugs. However, data for non-membrane-disruptive AMPs such as β-defensins are scarce, thus they still remain poorly understood. (2) We used single-cell fluorescence imaging to monitor the effects of a β-defensin (namely AvBD103b) in real time, on living *E. coli*, and at the physiological concentration of salts. (3) We obtained key parameters to dissect the mechanism of action. The cascade of events, inferred from our precise timing of membrane permeabilization effects, associated with the timing of bacterial growth arrest, differs significantly from the other antimicrobial compounds that we previously studied in the same physiological conditions. Moreover, the AvBD103b mechanism does not involve significant stereo-selective interaction with any chiral partner, at any step of the process. (4) The results are consistent with the suggestion that after penetrating the outer membrane and the cytoplasmic membrane, AvBD103b interacts non-specifically with a variety of polyanionic targets, leading indirectly to cell death.

## 1. Introduction

Antimicrobial peptides (AMPs), naturally produced by all living organisms, are key players of the innate immune system. They have been optimized during evolution leading to a vast diversity of molecular structures and mechanisms of action against a wide range of pathogenic microorganisms (bacteria, yeasts, fungi, etc.). AMPs have existed for millions of years without inducing widespread pathogenic resistance [1]. Indeed, only a few bacteria have naturally developed resistance mechanisms against AMPs [2,3,4]. One widely-advocated reason for this remarkable insensitivity to the buildup of resistance is that the main targets of AMPs are the pathogen’s envelopes. These complex structures cannot easily evolve without essential loss of function [2,5]. Moreover, growing evidence in the literature also shows that many AMPs disrupt several cellular processes simultaneously [6] and/or interact with numerous molecular targets (which can be either intracellular or membrane associated) rather than a single one [7]. This multiplicity of effects diminishes the likelihood of the emergence of a mutated pathogen able to resist all the combined mechanisms of action [8,9,10]. It is well known that due to the extensive use of antibiotics in the clinic, bacteria are able to develop resistance to almost all types of antibiotics (either by target modification, enzymatic degradation, and/or enhanced removal by efflux pumps [11,12,13] for review). For clinically used AMPs, such as polymyxin, colistin, or daptomycin, recent reports indicate the emergence of MDR (Multi-Drug Resistant) strains as well [11,14]. However, AMPs remain a promising alternative to conventional antibiotics, in particular in the fight against multi-resistant bacteria [15,16], and an increasing number of natural or engineered analogues of AMPs are currently in development, as reported on the last annual report of the World Health Organization on antibacterial agents [17].

Before rational engineering of highly effective compounds for clinical use (more potent, more rapid, less susceptible to proteases degradation, etc.), it is pivotal to first understand how AMPs work in nature. The ways in which AMPs can wreak havoc on bacteria can be broadly classified into two categories. The first category is characterized by direct and severe damage of the membrane, which rapidly leads to cell lysis and bacterial death. Such AMPs are called either “pore-former” [18], “lytic” [19], “membrane-damaging” [20], “membrane-active” [21], or “membrane-disruptive” peptides [11]. Several mechanisms of action have been proposed, mainly explored on model membranes [18,22], and particularly for linear helical AMPs. The second category presupposes rapid translocation of the AMP through the membrane without directly destroying it. It employs a wide variety of mechanisms, which have been scarcely studied and are therefore still poorly understood [21]. In such cases, interaction with bacterial cell surface remains an essential initial step of the global process of bacteria killing, but membranes are not the main targets [18,19,21,22,23,24]. Different intracellular molecules and physiological pathways can be targeted once the AMP reaches the cytoplasm, such as binding of DNA/RNA or inhibition of protein synthesis. Those AMPs are called “metabolic inhibitors” [18], “non-lytic” [25], “non-membrane permeabilizing” [25,26], or “non-membrane-disruptive” [11] peptides.

The term “defensin” covers two phylogenetically independent disulfide-rich peptide (DRP) superfamilies of AMPs [27]. They are generally cationic, and are expressed by a wide array of animals, plants, and fungi as essential components of their host defense system. Beta-defensins are a broad subfamily of defensins, present in all vertebrates, and thought to belong to the group of “non-lytic” AMPs. Among them, the king penguin β-defensin AvBD103b (Avian Beta-Defensin 103b, originally named spheniscin-2) is particularly appealing: contrary to most other β-defensins, where antibacterial activities are inhibited by salts (even physiological concentrations of salts), AvBD103b is remarkably salt-insensitive [28]. This unusual feature could be a considerable advantage for development of new drugs, in particular for treatments of infections of cystic fibrosis patients, where the airway surface liquid has high sodium chloride concentration [29] and for which there is an urgent therapeutic need [30].

AvBD103b was discovered in the stomach of the male penguin, which preserves food from attack by microorganisms at body temperature. This enables the male to feed the chick for its first 10 days of life [31]. AvBD103b displays antimicrobial activities against a large panel of microorganisms including Gram positive (Gram+) and Gram negative (Gram-) bacteria, yeast, and filamentous fungi [32]. Using NMR (Nuclear Magnetic Resonance), we determined its 3D structure (Figure 1A)—the first one obtained for an avian defensin [28]—which is comparable to those of mammalian β-defensins.

Only a few studies have attempted to dissect the mechanism of action of β-defensins [20,33,34,35,36,37,38,39,40,41,42]. They came to the conclusion that interaction with the membrane initiates an irreversible killing process, but is probably not the primary cause of death (See Appendix A for a comprehensive survey of the literature). The vast majority of these studies were performed at low salt concentration (typically 5–10 mM total salts), not at all representative of physiological conditions. Furthermore, despite their particular advantage for the development of new antibiotics, only two salt-insensitive β-defensins have been studied, AvBD103b [39] and the human β-defensin HBD3 [34,42]. The effects of AvBD103b on *Salmonella enteritidis*, have been examined; however, only part of the experiments were performed at the physiological salt concentration [39]. The main results from this preliminary study are in favor of a complex antibacterial mechanism that could not only damage the cell membrane but also interfere with intracellular DNA. This latter hypothesis has only been tested in vitro. Finally, all these data on β-defensins are not collected on live bacteria, which is essential to evaluate the real antimicrobial mechanism. The one exception is the recent work of Mathew and colleagues, who published the only real-time study of the action of four Human β-defensins (HBDs), including the salt-insensitive HBD3 [34]. These time-lapse fluorescence microscopy experiments were however performed at a non-physiological, low salt concentration. This study clearly demonstrated that HBDs do not kill *E. coli* by a simple mechanism involving membrane permeabilization, and that the membrane damages observed for the four HBDs do not directly correlate with their Minimal Inhibitory Concentration (MIC) values. Indeed, the authors propose that evolutionary selection has rendered some heterogeneity to bacterial killing mechanisms. It is therefore essential to go further in the study of these complex mechanisms and to establish efficient exploration methods on living bacteria under physiological conditions.

This paper describes fundamental research aimed at a more detailed understanding of the mechanism of action, an essential step for the design of optimized compounds and before any possibility of use in clinic. We monitor in real time and at a physiological concentration of salts the effects of the King penguin β-defensin AvBD103b on living *E. coli*. We use single-cell fluorescence imaging, following the method we previously developed for membrane-active linear helical AMPs [43,44,45]. Single-cell microscopy is extremely powerful in precisely determining: (i) the time to permeabilization of the outer membrane (OM), if it occurs; (ii) the time to permeabilization of the cytoplasmic membrane (CM), if it occurs; and (iii) the effect on bacterial growth (halting of growth, cell shrinkage, etc.). These elements are key parameters to dissect the mechanism(s) of action of AvBD103b. The cascade of events we recorded suggests an original mechanism, significantly different from all the other antimicrobial compounds that we previously studied using the same technique and under the same physiological conditions. Finally, we applied the same approach to the all-D version of AvBD103b, in order to scrutinize if any step in the mechanism could be influenced by a stereospecific interaction.

## 2. Results and Discussion

### 2.1. Evaluation of the Minimal Inhibitory Concentrations

The lowest concentration for which no cell growth is detectable in the media used for fluorescence spectroscopy (i.e., EZRDM + ampicillin, see Section 3.2), considered as a good estimation of the Minimal Inhibitory Concentration (MIC), was 4 μM for L-AvBD103b. The same value was measured for the control antimicrobial cathelicidin LL-37, in accordance with our previous works [43,46,47]. For D-AvBD103b, the mirror image of the natural L-AvBD103b, the MIC was estimated at 8 μM, which cannot be considered significantly different from the 4 μM observed for the L-isomer.

### 2.2. Delayed and Transient Effect of L-AvBD103b Defensin on E. coli Membrane Permeability

We followed, in real time, on living *E. coli*, the effect of L-AvBD103b on OM and CM permeabilization, over a one-hour movie with interleaved GFP, Sytox and phase contrast images. The *E. coli* strain JCW10 (see Section 3.2) expresses and exports the fluorescent GFP to the periplasm, the thin space between the OM and CM. At time *t* < 0 (i.e., before the onset of flow of peptide), a halo of green fluorescence was characterized by a double-peaked transverse intensity profile from 488 nm channel (Figure 1B). This indicates that, as desired, most of the fluorescent GFP resides in the periplasm. At this time *t* < 0, the nucleic acid-binding Sytox dye does not induce any intracellular fluorescence, proving that it could not reach the intracellular DNA/RNA and confirming the membrane integrity of the bacteria.

Most experiments were carried out at 4 μM. Additional experiments were carried out at 2 and 8 μM to evaluate concentration effects. The sequence of events detailed below (i.e., osmotic effect and halting of growth during Phase 1, abrupt transient permeabilization of OM and CM, and slow leakage of dyes across the cell envelope during Phase 2 is similar for the three concentrations, but each event tends to occur earlier with the highest peptide concentration. One typical example is illustrated in Figure 1.

#### 2.2.1. Phase 1: Apparent Osmotic Effects and the Halting of Growth as Defensin Accesses Periplasm

We define Phase 1 of the attack as the time window during which GFP remains localized in the periplasm. The first observable change is a redistribution of the periplasmic GFP fluorescence from a narrow halo image (Figure 1B) into an image with much of the intensity concentrated at one or both of the endcaps (Figure 1C). Such features are observed in only 10–50% of the cells on a given plate, regardless of the peptide concentration. At 4 μM of L-AvBD103, the bright cell tips usually appear 1–3 min after the onset of flow of peptide, last less than 3 min, and then dissipate. Similar transient features were previously observed in a study of the attack of the cationic β-peptide-based nylon-3 random copolymer MM_63_:CHx_37_ on *E. coli* [48], and we called them “periplasmic bubbles”. They are highly reminiscent of the GFP images resulting from a well-studied perturbation known as plasmolysis, in which an external osmotic upshift withdraws water from the cytoplasm. To accommodate the decrease in cytoplasmic volume while preserving the surface area of the CM, one or both of the CM endcaps invert, in effect transferring the endcap volume to the periplasm to form “plasmolysis spaces”. In the present case of perturbation by L-AvBD103b, we suggest that the rapid translocation of the cationic peptide across the OM and into the periplasm acts similarly to an external osmotic upshift. The resulting osmotic imbalance between cytoplasm to periplasm causes a transfer of water from cytoplasm to periplasm to rebalance [49]. The endcaps of the CM collapse while the OM retains its original shape. Periplasmic GFP pools in the enlarged periplasmic endcaps (Figure 1C). The total integrated GFP fluorescence intensity is globally constant during this Phase 1, indicating that the 4.5 kDa L-AvBD103b has crossed the OM without permeabilizing it to the 27 kDa GFP. However, at this stage, we cannot rule out some leakage of periplasmic GFP into the surroundings, due to the imperfect baseline subtraction.

Towards the end of Phase 1, the growth of the bacterial cell halts completely (Figure 1D, black line), as judged by cell length measured from the phase contrast images, and remains constant for the rest of the experiment, and regardless of the peptide concentration. This halting of growth occurs abruptly, except in a few cases. Most commonly at the lowest peptide concentration, bacterial growth slows down more gradually before stopping completely. The halting of growth is not correlated in time with the formation/dissipation of periplasmic bubbles. It always occurs after the dissipation of the bubbles, and it occurs even when formation of the bubbles is not observed. 

#### 2.2.2. Abrupt Transition Time *t*_1_: Strong, Transient Permeabilization of OM and CM

We defined *t*_1_ (Figure 1D) as the time at which the total GFP fluorescence intensity abruptly decreases by 10–20% in one or two camera frames (12–24 s) (Figure 1D, green curve). This occurs whether or not periplasmic bubbles have been observed during Phase 1. Partial loss of total GFP intensity indicates that the peptide has caused the OM briefly to become readily permeable to GFP, so that part of the GFP is lost to the cell surroundings. At the same time, the spatial distribution of GFP fluorescence abruptly changes from a predominantly periplasmic halo to that of a predominantly filled cytoplasm (Figure 2). This pattern partly reflects the residual presence of GFP in the cytoplasm, where it is produced, and which was previously masked by the strong periplasmic GFP signal (Figure 2A). In some cases, the intensity of the cytoplasmic GFP signal clearly increases, showing that part of the periplasmic GFP accesses the cytoplasm through the CM (Figure 2B). This is shown most clearly by an increase in the central GFP peak in transverse intensity linescans.

Concomitantly with the abrupt loss of GFP at time *t*_1_, Sytox fluorescence always begins to appear (Figure 1D, orange curve), whatever the concentration of peptide used. The concentration of L-AvBD103b affects both the fraction of cells that exhibits such an abrupt loss of GFP and the distribution of times *t*_1_ at which the events occur. At 2 μM of L-AvBD103b, only ~50% of the bacteria exhibit the abrupt loss of GFP during the 60 min movie. For 10 such bacteria, the mean value <*t*_1_> was 25 min, with a range of 16–32 min. At 4 μM, 40 to 90% of cells on a given plate showed the same behavior; for 19 such events, we found that mean value <*t*_1_> was 10 min, with a range of 5–20 min. There is no significant difference between 4 and 8 μM.

#### 2.2.3. Phase 2: Slow Leakage of GFP and Sytox across the Cell Envelope

After the abrupt leakage event at *t*_1_, which marks the beginning of Phase 2, the rate of loss of GFP intensity quickly decreases. As reported in Figure 1D (green curve), the cell continues to leak GFP to the surroundings, but more slowly for the remaining 30 min of the movie. In this transient permeabilization event [50,51], the peptide must attack the cell layer by layer from the outside in, meaning OM before CM, and each outer leaflet before each inner leaflet. The binding of more and more peptide molecules to the outer leaflet of a bilayer causes mechanical stress, eventually leading to local “bursting” events. When massive enough, these events enable transit of the 27 kDa GFP, of the small Sytox dye (approx. 0.5 kDa) and of the 4.5 kDa peptide itself, across the bilayer. The peptide now has access to the inner leaflet of the same bilayer. As the peptide concentration rises up on the inner leaflet, the mechanical imbalance is alleviated, and the membrane can reform a bilayer that is more resistant to GFP passage. However, during this time the much smaller Sytox dye continues to leak into the cytoplasm from the cell surround. Evidently the cell envelope is only partially healed, greatly impeding GFP transit but enabling Sytox transit. Similar transient permeabilization and healing behavior was previously observed for both the venom component melittin [45] and the above-mentioned cationic polymer MM_63_:CHx_37_ [48]. On the contrary, other AMPs such as LL-37 [43] or cecropin A [52] released periplasmic GFP completely to the cell surroundings.

We can monitor the progress of the L-AvBD103b peptide within the bacterial cell following GFP and Sytox fluorescence only indirectly. We infer that the peptide has accessed the cytoplasm because: (i) After the abrupt transition at *t*_1_, the peptide-induced membrane defects are clearly sufficient to allow some of the periplasmic GFP to partially leak through the OM and dissipate into the external cellular environment (Figure 1D, green curve), and/or through the CM and into the cytoplasm (Figure 2B); (ii) From time *t*_1_, a weak fluorescence signal from Sytox begins to rise (Figure 1D, orange curve), indicating that the DNA stain has traversed both OM and CM and gained access to the cytoplasm, where it can bind to the chromosomal DNA and become fluorescent. The Sytox signal continues to rise slowly to the end of the 60 min movie, corroborating the residual permeability of both OM and CM to the dye. Eventually, all bacteria are labeled with Sytox (Figure 3), even those cells which did not exhibit abrupt GFP loss. The slow kinetics observed here for *E. coli* is in agreement with the previous work performed on *S. enteritidis* by Teng and Coll [39]. Indeed, they showed by flow cytometry experiments that only 39% of the *S. enteritidis* treated with L-AvBD103b have lost their integrity after 30 min, and are stained by propidium iodide. Non-lytic AMPs are known to cause a relatively slow process of cell death, in contrast with the lytic ones that cause rapid cell death by direct membrane permeabilization and immediate loss of intracellular components [53,54,55]. The process described above is slow, and after 60 min under L-AvBD103b, most of the cell envelopes appear intact (see exceptions below). This slow evolution of events is clearly not in favor of a direct membrane-disruptive mechanism of killing.

It seems likely that defensin gains access to the cytoplasm and disrupts essential physiological processes, ultimately leading to the death of the bacteria. What molecules can AvBD103b interact with? The cytoplasm is a dense cellular compartment [56] filled with negatively charged biomolecules (e.g., DNA, RNA, or some proteins) whose charge is offset by small cations such as K^+^ and Mg^2+^ [57]. Any cationic AMP able to enter the cytoplasm will automatically disturb this fine internal electrostatic balance, and induce deleterious effects [47]. Indeed, a cationic AMP could interact—at least electrostatically, and even without any specificity—with any of these internal negatively charged targets, disturbing their normal functions. 

### 2.3. Final Indirect Effect of L-AvBD103b on E. coli

Most bacteria appear intact at the end of the record (50–55 min of recording after injection of the peptide), without visible morphological changes. A small proportion of the cell envelopes have “exploded” during this time. These observations are reminiscent of TEM images recorded for various bacteria (Gram- *S. enteritidis*, *S. typhimurium*, *M. catarrhalis*, or Gram+ *C. perfringens*, *S. pneumonia)* treated with different vertebrate β-defensins, resulting in membrane leakage, loss of cytoplasmic content, and lysis at the septum of dividing cells [33,39,40,41,42]. However, to our knowledge this is the first observation of such an explosion of bacteria in real time by fluorescence microscopy. Explosions are rare events, but we focus on the bacteria that exploded, as this event reflects some peptide-induced fragility.

When an explosion occurs, the cytoplasmic content is suddenly (i.e., in less than our 12 s interval between two images) ejected from the cell, as visualized in the phase contrast images (Figure 4, left column). Sytox fluorescence flashes around the exploded cells (Figure 4, middle column), showing that the chromosomal DNA is rapidly exposed to the Sytox molecules present in the surrounding medium. Finally, GFP fluorescence dramatically decreases or totally disappears within these 12 s (Figure 4, right column). 

These rare explosions are difficult to record in real time given our observation time and considering that the field of view contains only 5 to 20 bacteria. An example of explosion occurring at time *t* = 49 min is reported in Figure 4 and in the corresponding video, available as Appendix A. In order to roughly evaluate the number of explosions occurring within 1 h, additional fields of view were quickly checked just after main video acquisition (between time *t* = 55 and 60 min after injection of peptide). Roughly 5–10% of bacteria have exploded within 1 h at 4 μM of L-AvBD103b. In an attempt to catch more explosions, we increased the concentration of peptide or the video time. Thus, we performed three additional experiments at 8 μM and two additional longer (90 min) experiments at 4 μM of peptide. Unfortunately, at 8 μM, we did not observe any explosion within 1 h in the selected field of view. In contrast, more explosions occurred in longer experiments at 4 μM, between 60 and 90 min, which confirmed the relatively slowness of the process.

The explosions always occur long after the apparent halting of growth. What could lead to explosions? If the cell envelope is somehow weakened, the internal pressure of the cytoplasm, normally counterbalanced by the pepdidoglycan layer, can no longer be contained. For example, AvBD103b could interfere with pepdidoglycan biosynthesis, as described for other AMPs through various mechanisms: (i) as proposed above to explain the halting of growth during Phase 1, AvBD103b could directly bind to peptidoglycan, like LL-37 does [43]; (ii) AvBD103b could bind one of the peptidoglycan precursors, as do some defensins from invertebrates [58,59,60] or other antimicrobials [61] with the essential precursor lipid II [62]; (iii) another possibility is that AvBD103b could interfere with fluid lipid micro-domains of the CM, then delocalizing essential peripheral membrane proteins, such as enzymes involved in pepdidoglycan biosynthesis. This phenomenon has been described for the hexapeptide RWRWRW [63] and for the cyclic lipopeptide daptomycin [64], which disturb the lipid II synthase MurG. Finally, we cannot exclude the weakening of the cell envelope by any other mechanism, independent of the peptidoglycan.

### 2.4. Antibacterial Mechanism of AvBD103b Does Not Involve Significant Stereo-Selective Interactions with Any Partner

D-peptides are “mirror images” of the natural L-peptides, and composed of the inverted-chirality D-enantiomers of the proteogenic L-amino acids. They have long been used as a tool to probe the molecular mode of action of bioactive peptides, in order to discriminate membrane- or receptor-mediated modes of action [65]. If the interaction occurs with achiral targets, then the D- and L-peptides bind in an intrinsically indistinguishable manner. Otherwise, if the interaction occurs with targets with high chirality, such as a protein receptor, the mirror image (D-) of the specific natural L-peptide usually has no affinity. These unique features have been particularly used in the study of AMPs. In a pioneering work in 1990, Merrifield and colleagues reported that the D- and L-versions of several linear cationic AMPs (cecropin, melittin and magainin) have similar MIC on a series of Gram- and Gram+ strains [66], suggesting that their molecular mode of action relies on a direct interaction with the assumed-achiral bacterial lipid membrane. In strong contrast, the mirror images of the fungal defensin plectasin [67], of proline-rich antimicrobial peptides (PrAMPs) [68,69], or of the insect β-haipin thanatin [70] lost their antibacterial activity. Highly specific and high affinity binding to lipid II is the main effector of plectasin’s capabilities [59], whereas PrAMPs have been shown to bind specifically to bacterial ribosomal proteins and inhibit protein synthesis [71,72,73]. Thanatin was first shown to bind lipopolysaccharide (LPS) in vitro [74]. Further studies proposed the inhibition/disruption of LPS transport complex assembly as its main killing mechanism against Gram-negative bacteria [75,76].

However, this binary vision “chiral partner vs. achiral partner” is not as clear-cut as previously thought. Indeed, the paradigm of achiral bacterial lipid membrane has been revisited, showing that the “slightly” chiral environment of lipid bilayers arising from the chirality of glycerol in phosphoglycerolipids can modulate the function of membrane-active peptides and challenges the view that peptide−lipid interactions are achiral [77,78]. Moreover, non-stereospecific interactions can occur with highly chiral partners, as exemplified in the case of the bacterial non-ribosomal lipopeptide tridecaptin A1 (TriA_1_), where D- and L-TriA_1_ bind to *E. coli* LPS with similar affinity [79]. Then, comparable MICs do not systematically signify an exclusive mechanism of membrane permeabilization, but may reflect any non-specific interaction with any (chiral) component of the bacterial cell envelope or with intracellular target(s), such as DNA. For example, the most studied linear cationic AMP, LL-37, known to permeabilize membranes without stereospecificity (equal MIC for L- and D-form), has been shown to interact with negatively charged cell wall components such as LPS or lipoteichoic acid (LTA) [80]. More recently, it has been proposed that after permeabilization, a portion of LL-37 could enter the cytoplasm and electrostatically bind to RNA, DNA, and/or ribosomes [47,81].

Many AMPs employ multifaceted bacterial killing mechanisms that may act in concert, and each mechanism could be stereo- and/or non-stereospecific. Comparison of the D- and L-isomers is essential to discriminate the importance of the different molecular recognition events responsible for the overall activity. Therefore, we found it particularly relevant to apply our AMP-induced bacterial death video monitoring approach to the D-version of AvBD103b, in order to scrutinize if any of the above-defined steps in the bacteria-killing mechanism of AvBD103b could be influenced by a stereospecific interaction. Before our study, very few data were available in the literature to compare L- and D-isomers of β-defensins, the subgroup to which AvBD103b belongs. Comparable MICs were measured on a series of Gram- and Gram+ bacterial strains for L- and D-isomers of the human HBD2 [82] and the avian AvDB2 [83]. Similarly, we obtained comparable MICs values against *E. coli* for L-AvBD103b and D-AvBD103b.

Our real-time experiments in living bacteria showed that every step of the cascade of events, i.e., formation of periplasmic bubbles, halt of growth, abrupt and partial loss of GFP, entry of Sytox, and explosions, is the same for D-AVBD103b as for L-AvBD103b (see details in Appendix A). Moreover, occurrence frequencies and the timing of events are also comparable. It is thus clear that the antimicrobial mechanism of AvBD103b does not involve significant stereo-selective interaction with any partner, at any step of the process. If our preferred hypothesis is that (part of) AvBD103b could be able to access the cytoplasm, and to interact non-stereo specifically with one or more intracellular partners, we do not exclude a non-stereospecific interaction with *E. coli* cell wall components such as peptidoglycan or LPS layers. Undoubtedly, AvBD103b may be capable of disrupting multiple processes essential for the bacteria, such as sand in a gearbox, and ultimately leading to its death.

### 2.5. The Permeabilization Profile of AvBD103b Is Atypical 

There are multiple ways for AMPs to wreak havoc on bacteria, and nature provides a useful diversification of mechanisms during evolution. With the same technique (fluorescence microscopy in real time on living bacteria) and the same protocol (bacterial strain, buffer, pH, temperature, dyes, etc.) we previously video-monitored the permeabilization of several other antimicrobial compounds: the most studied linear AMPs, auto-organizing in amphiphilic helices in contact to the bacterial membrane (cathelicidin LL-37, melittin, and cecropin A) and of the synthetic cationic MM_63_:CHx_37_ polymer. Thus, we are able to make a detailed and reliable comparison of each step of the mechanisms of action. In fact, each agent appears to have its own characteristics (Table 1).

Human LL-37 induces an abrupt loss of periplasmic GFP, with a greater fractional loss than measured for AvBD103b, reflecting greater damage of the OM [43]. Contrary to AvBD103b, the increase in Sytox fluorescence is more abrupt, and does not correlate in time with the loss of GFP (abrupt loss of periplasmic GFP at *t* = 9 min, and abrupt permeabilization of Sytox at *t* = 16 min, on average at 1x MIC). Cell shrinkage correlates with the abrupt loss of periplasmic GFP and may reflect the interference of LL-37 with the peptidoglycan layer, as soon as it reaches the periplasm. For AvBD103b, halting of growth mainly occurs before the abrupt (but partial) loss of periplasmic GFP, which suggests that AvBD103b is able to reach the periplasm without extensively disturbing the OM. Contrary to AvBD103b, no periplasmic GFP bubbles were seen with LL-37, perhaps because the rapid loss of GFP to the cell surroundings precludes their observation.

Cecropin A, a 37-residue linear helical peptide from moths, induces cell shrinkage after two minutes, and drastically permeabilize OM at the septal region at the same time. CM permeabilization always follows the OM permeabilization (2–3 min latter), but occurs at one endcap [84]. While the leakage of periplasmic GFP outward through the OM is abrupt (less than 1 min), the leakage of Sytox into the cytoplasm through the CM occurs over more than 20 min [52]. Cecropin A behavior is thus significantly different from the AVBD103b cascade of events.

In the case of melittin, a 26-residue linear helical peptide from bee venom, transient periplasmic bubbles are observed. However, contrary to AvBD103b, these periplasmic bubbles last less than 24 s, occur at the septum, and quickly leak into the large cytoplasmic volume and/or outside. At the same time, within the first two minutes of experiment (at 1x MIC), the GFP fluorescence abruptly decreases (a loss of around 50% of GFP, i.e., more than with AvBD103b), cells shrink and a strong Sytox fluorescence starts [45]. Melittin permeabilizes the membrane highly efficiently. However, we reported a surprising sequence of permeabilization, re-sealing, and re-permeabilization of both OM and CM, presumably related to curvature stress (wedge effect). An analogous, partial membrane-healing effect may occur for AvBD103b attack. This may explain the slowing of GFP loss after CM permeabilization (at *t*_1_) for some cells (See Appendix A).

In a very interesting way, some of the events reported for the synthetic copolymer MM_63_:CHx_37_ [48] mimic the AvBD103b behavior. Under the effect of MM_63_:CHx_37_, *E. coli* first shrinks and exhibits periplasmic bubbles, within seconds. The copolymer induces the formation of periplamic bubbles at both endcaps, which are very similar to those created with AvBD103b. The copolymer then induces inward movement of periplasmic GFP, which indicates: (i) its ability to permeabilize the CM, and (ii) its translocation across the OM without complete permeabilization to GFP. In fact, unlike natural linear AMPs, the copolymer does not permeabilize the OM to GFP, over a 50 min observation period. The GFP spatial distribution changes in 12–24 s from a periplasmic distribution to a cytoplasmic distribution, which is very similar to our observations for AvBD103b (Figure 2B and Appendix A). However, for the copolymer, after shrinking during the first seconds, cell length then recovers until the time at which the CM is permeabilized to GFP. This is very different from our observations for AvBD103b.

## 3. Materials and Methods

### 3.1. Defensins Synthesis

AvBD103b was synthesized by Fmoc-based Solid Phase Peptide Synthesis (SPPS) and oxidatively folded as described previously for AvBD2, a β-defensin from the chicken Gallus gallus [83]. The primary structure is SFGLCRLRRGFCARGRCRFPSIPIGRCSRFVQCCRRVW, with cysteinyl residues involved in the C1–C5, C2–C4, C3–C6 disulfide bridges array. Its enantiomer D-AvBD103b was synthesized using the same protocol using D-amino acids building blocks (See Appendix A).

### 3.2. Bacterial Strain, Cultures Conditions

The *Escherichia coli* strain JCW10, derived from *E. coli* K12 (MG1655), contains the previously described pJW1 plasmid [86], which has ampicillin resistance and a tetracycline-inducible promoter and that expresses the TorA-GFP fusion protein, composed of the twin-arginine signal peptide of TMAO reductase (TorA) and the 27 kDa green fluorescent protein (GFP). On transport to the periplasm, the 10–50 nm thick space between CM and OM, the sequence signal is cleaved and the fluorescent signal of the fully folded, mature periplasmic GFP can then be imaged [87]. Bacterial cultures were grown overnight at 30 °C from a glycerol frozen stock solution to stationary phase. All bacterial cultures were grown in a low-fluorescence chemically defined EZRDM medium (EZ Rich Defined medium). EZRDM is a MOPS-buffered solution at pH 7.4 (Teknova 2137, Teknova, CA, USA), supplemented with 2 mg/mL glucose, 1.32 mM K_2_HPO_4_, 76 mM NaCl, nucleic acids (Teknova 2103, Teknova, CA, USA) and vitamins (Tennova 2104, Teknova, CA, USA). Ampicillin (100 μM) was added to all buffers.

### 3.3. Evaluation of the Minimal Inhibitory Concentration

Bacterial subcultures were grown at 30 °C from the *E. coli* JCW10 overnight culture to exponential phase. Serial dilutions were performed for each peptide to estimate the MICs by the broth dilution method on a 96-well plate, each plate containing *E. coli* JCW10 (A_600_ = 0.05) in the EZRDM medium described below. After 6 h at 30 °C, the lowest concentration for which no cell growth is detectable at 595 nm is considered as the MIC. The antimicrobial peptide cathelicidin LL-37 (61302, Anaspec, CA, USA) was used as a positive control.

### 3.4. Preparation of PDMS Microfluidic Devices

Microfluidic patterned polydimethylsiloxane (PDMS) devices were prepared as described in [88] with Sylgard 184 silicone elastomer mixture. The final PDMS-based microfluidic chambers consist of a single rectilinear channel of uniform height of 150 μm, width of 6 mm and length of 11 mm. The chamber volume is 10 μL. To ensure optimal bacterial adhesion, the device is filled with 10 μL of an aqueous solution of 0.01% of poly-L-lysine (molecular weight > 150,000 Da) and kept overnight at 30 °C.

### 3.5. Fluorescence Microscopy

For imaging, subcultures were grown at 30°C from the overnight culture (see Section 3.2) to the exponential phase. A 3 mL subculture (1:100 dilution from the overnight culture) was induced after 75 min (A_600_ about 0.1) with 2.25 μL of a 0.05 mg/mL tetracycline solution (37.5 μg/L final concentration) for 4 min, according to the previously described protocol [45]. After an additional 75–90 min at 30 °C, bacteria were ready to image, i.e., the A_600_ was between 0.3 and 0.5 and GFP expression was mainly periplasmic. Bacteria were injected into the pre-warmed PDMS device (pre-installed at least 30 min on the microscope, at 30 °C), straight from the subculture. After several minutes, all non-adhered cells were rinsed away with 1 mL of EZRDM. The plated cells grew normally. Efficient expression and subcellular localization of the GFP to the periplasm was checked using transverse intensity line scans. The culture was not synchronized so that all phases of the cell cycles were present simultaneously.

All images were acquired on a Nikon Eclipse Ti inverted microscope with an oil immersion 100×, 1.45 N.A. phase contrast objective (CFI Plan Apo Lambda DM; Nikon Instrument). The images were further magnified 1.5×. All images were recorded by a back-illuminated EMCCD camera with 16 μm × 16 μm pixels (Andor iXon DV-897). Each pixel corresponds to 105 × 105 nm^2^ at the sample with an overall magnification of 150×. During typical 60 min movies, the imaging sequence cycle (488 nm excitation, 561 nm excitation, and phase contrast illumination) repeated every 12 s, with an exposure time of 50 ms for each exposure. Fast shutters (Uniblitz LS2; Vincent Associates) synchronize illumination and image acquisition. The µManager open source software [89] was used to record the data and to switch filters between frames, using a LB10-NW filter wheel (Sutter). To minimize spectral bleed-through in the two-color experiments, we used the narrower filters HQ510/20 and HQ600/50M for the green and red channel, respectively. Laser intensities at the sample were typically ~5 W/cm^2^ and ~2.5 W/cm^2^ at 488 and 561 nm, respectively. Interleaved observations by phase contrast microscopy allowed measurement of cell length vs. time, a proxy for cell growth.

A typical field of view contains 5 to 20 adherent bacteria. They were imaged with three channels (interleaved GFP, Sytox orange and phase contrast images) for 5–10 min before the onset of flow of peptide in order to check the expression and localization of GFP, the absence of red background fluorescence in the 561 nm channel, and the normal growth of bacteria, respectively. Time *t* = 0 of the movie corresponds to the beginning of exposure to a flow of β-defensin (at 0.5 to 2 × MIC) in EZRDM/ampicillin, in the presence of 10 nM of Sytox orange (S11368, Thermo-Fisher Scientific). During the first minute, the flow of peptide was set to 0.2 mL/min, then fixed at 0.2 mL/h for the rest of the experiment. Generally, a one-hour movie with 300 cycles (frames) was acquired to follow the events in real time. The system was kept at 30 °C during the whole experiment.

Experiments were carried out at 2 μM, 4 μM, and 8 μM for L-AvBD103b, and at 4 μM and 8 μM for D-AvBD103b. Control experiments were performed in the exact same conditions, but without peptides, to verify the absence of effects of all other components, and the inability of Sytox to enter the cytoplasm in the absence of peptide.

## 4. Conclusions

A general strength of AMPs as antimicrobial agents is the variety of their mechanisms of action, provided and optimized over millions of years by nature. However, these mechanisms remain poorly understood. Understanding how AMPs work in nature is the first essential step towards rational engineering of highly effective compounds for clinical use, to be used by themselves or in combination. The identification of atypical mechanisms, unlikely to rapidly induce resistance, is urgently needed for the development of original/atypical ways to kill bacteria.

Very few studies have attempted to dissect the mechanism of action of β-defensins and none of these studies were performed under our conditions, i.e., live targets, real time, and physiological salt conditions. For the first time, using single-cell fluorescence imaging, we video-monitored the effects of the AvBD103b β-defensin from the king penguin on living *E. coli,* and at physiological salt conditions. Our results favor a non-membrane-disruptive, non-specific, and likely multifaceted mechanism of action. The cascade of events, inferred from our precise timing of membrane permeabilization effects, associated with the timing and observation of bacterial growth arrest, differs significantly from other antimicrobial compounds we have previously studied in the same experimental conditions. Our data indicate that AvBD103b would be able to reach intracellular targets such as DNA and/or disrupt the integrity of the bacterial envelope. Importantly, the cascade of events is the same for D- and L-isomers, supporting the fact that the AvBD103b mechanism does not involve a significant stereoselective interaction with any partner, at any step of the process. AvBD103b most likely interacts non-specifically with a variety of polyaniomic targets. Our hypothesis is that AvBD103b is able to access the cytoplasm, and to interact non-stereo specifically with one or more intracellular partners. However, we do not exclude a non-stereospecific interaction with the *E. coli* cell wall components such as peptidoglycan or LPS layers. AvBD103b may be capable of disrupting multiple processes essential for the bacteria, like sand in a gearbox. The resulting additive detrimental effects would prevent bacteria from developing a resistance mechanism, which is an undeniable advantage in the development of new antibiotics.

Finally, our D- vs. L-real-time microscopy bactericidal activity-dissecting approach is unprecedented, and such a powerful tool could be applied to other AMPs in order to better characterize their mode of action, by assessing the stereospecificity of each step of the bacteria killing process.

## Figures and Tables

**Figure 1 ijms-23-02057-f001:**
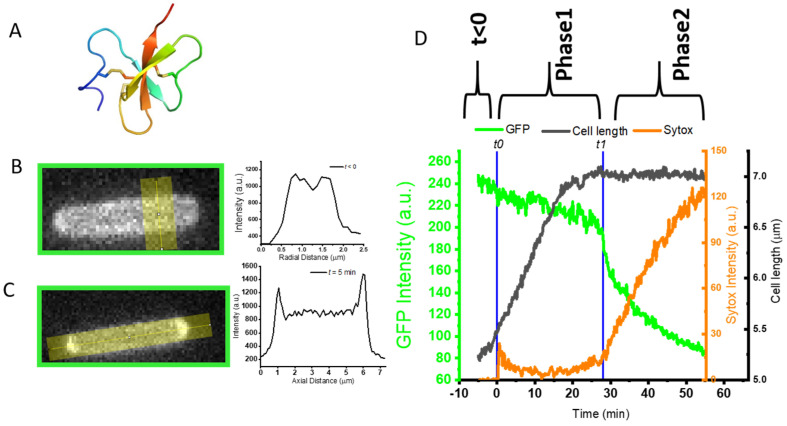
Effects of 2 μM of L-AvBD103b on a representative *E.coli* JCW10 strain expressing periplasmic GFP. (**A**) Schematic representation of the 3D NMR structure of L-AvBD103b [28]; PDB code 1ut3; backbone rainbow colored from N-ter in blue to C-ter in red; disulfide bridges in yellow; (**B**) At time < 0, typical fluorescence halo and typical double-peaked transverse intensity profile along the yellow line, indicating that most of GFP is in the periplasmic space before adding peptide; (**C**) During Phase 1, typical bright transient area of green fluorescence at end caps of *E. coli*, indicating the formation of transient periplasmic bubbles, for this specific bacteria from *t* = 2.4 min to *t* = 8 min; (**D**) Time dependence of total GFP intensity (in green, arbitrary units), of total Sytox intensity (in orange, arbitrary units) and of cell length (in black, in μm, calculated from phase contrast image). The flow of peptide begins at *t* = 0. Abrupt transition at time *t1* (28 min for this example) is highlighted in blue.

**Figure 2 ijms-23-02057-f002:**
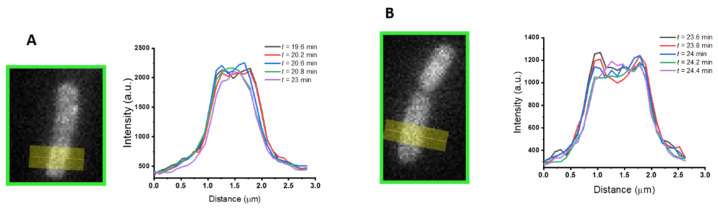
Examples of GFP fluorescence profile evolution around time *t*_1_. The transverse intensity linescans (arbitrary units), following the yellow line, were reported at different times *t* (in min) around *t*_1_. Two typical bacteria illustrate: (**A**) GFP lost to the cell surrounding through OM; and/or (**B**) access to the cytoplasm through the CM.

**Figure 3 ijms-23-02057-f003:**
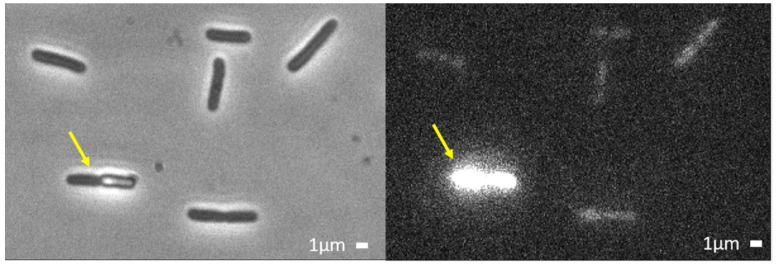
Example of Phase contrast image (**left**) and of Sytox fluorescence (**right**) after 1 h flow of 4 μM L-AvBD103b. All bacteria are Sytox-labelled, generally very weakly. In a minority of cases, the high Sytox fluorescence testifies that both OM and CM were more strongly permeabilized (one strongly labelled bacteria in this example, yellow arrow).

**Figure 4 ijms-23-02057-f004:**
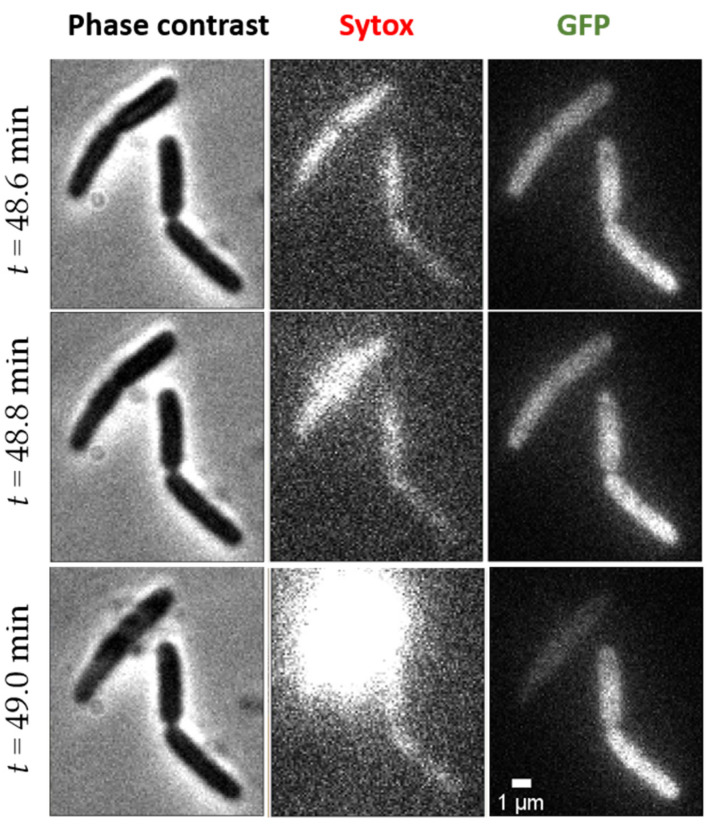
Example of *E. coli* explosion occurring at time *t* = 49 min of exposure to 4 μM L-AvBD103b. Phase contrast (**left**), Sytox (**middle**), and GFP (**right**) images are represented at time 48.6 min (top, before explosion), at time 48.8 min (middle, very early sytox explosion), and at time 49 min (bottom, final state).

**Table 1 ijms-23-02057-t001:** Comparison of L- and D-AvBD103b with other antimicrobial compounds, which we previously studied with the same technique (real-time fluorescence video microscopy, on living bacteria) and under the exact same conditions (30 °C, EZRDM buffer solution at pH 7.4).

Antimicrobial Compound (Charge, Hydrophobicity)	Observation of Periplasmic Bubbles (Duration)	Event1: OM Permeabilization (GFP Leak to Surroundings)	Event2: CM Permeabilization (Sytox Increase)	Concomitance of Event 1 and Event 2	Halt of Growth or Cell Shrinkage and Correlation with Event 1	Refs
LL-37 (+6, 35%)	No	Very High	High	No	Cell shrinkage correlates with event 1	[43]
Cecropin A (+6, 48%)	No	Very High	Weak	No	Cell shrinkage correlates with event 1	[52,84]
Melittin (+6, 46%)	Yes (<24 s)	High	High	Yes	Cell shrinkage correlates with event 1	[45]
MM_63_:CHx_37_ (mean charge +22)	Yes	Weak	High	No	Cell shrinkage precedes event 1	[48,85]
L-AvBD103b (+10, 47%)	Yes (minutes)	Weak	Weak	Yes	Halt of growth precedes event 1	This work
D-AvBD103b (+10, 47%)	Yes (minutes)	Weak	Weak	Yes	Halt of growth precedes event 1	This work

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
