# Peer review of "Real-Time Fluorescence Microscopy on Living E. coli Sheds New Light on the Antibacterial Effects of the King Penguin β-Defensin AvBD103b"

_ijms, 2022, doi:10.3390/ijms23042057_

Round 1
Reviewer 1 Report
Comments are attached in a separate file.

Reviewer 2 Report
The study by Landon et al. 2022 reports an innovative approach to shed light over the possible mechanisms of defensin-like peptides on bacteria. The study has some ups and downs, which are described below. After carefully evaluating the manuscript, I suggest major revisions before publication in IJMS.
Abstract:
Lines 13 – 14 “which are considered unlikely to induce bacterial re- 13 sistance, are a promising alternative to conventional antibiotics”. Please rephrase. A growing number of studies have reported the induction of bacterial resistance to AMPs using sub-lethal concentrations (mainly OMICS studies).
Please clarify in the abstract whether this defensin interacts with intracellular targets and, as a secondary mechanism, disrupts the bacterial membrane or not. It is unclear what “low-affinity targets” are.
Introduction:
Lines 34 to 38: “AMPs have existed for millions 34 of years without inducing pathogen resistance [1]. One widely-advocated reason for this 35 remarkable insensitivity to the buildup of resistance is that the main targets of most AMPs 36 are the pathogen’s envelopes. Indeed, these complex structures cannot easily evolve with- 37 out essential loss of functions [2]” This is also questionable. As mentioned above, an increasing number of studies have reported bacterial resistance do AMP, mostly those that act through membrane-associated mechanisms. Contrary to what the authors claim, proteomics and transcriptomics studies have revealed that bacterial resistance to AMPs commonly involve modifications in lipid metabolism and, therefore, the phospholipids composition and proportion at the bacterial envelope (thus avoiding AMP’s attachment and reorientation). Please rephrase, update these concepts and some works showing that bacterial resistance to AMPs, in fact, occurs (naturally or induced).
Lines 42 to 44: “Since they are considered unlikely 42 to induce bacterial resistance [5], AMPs are a promising alternative to conventional anti- 43 biotics, in particular in the fight against multi-resistant bacteria [6, 7].” The same applies here. Please update accordingly.
Line 46: WHO ïƒ please provide the full name before the abbreviations.
For a more reliable comparison between the different defensins presented (SI) and their activities, please provide all concentrations in micromolar.
The quality of figures 1B and C is bad. The same for the labels in the attached graphs. Please double-check
In figure 1D the colors for GFP, SYTOX and length are also difficult to correlate to the lines within the graph. Please correct it accordingly.
Check italics for species names throughout the main text and figure caption.
Please adjust figure 2. The graphs’ axis has no units and, sometimes, labels are missing. It difficult the interpretation
Figure 3 lacks quality and labels as well (arrows, etc.).
Up to the topic 2.2.3 the data presentation is quite confusing. The authors infer different levels of peptide translocation across OM and CM by measuring GFP and SYTOX. All these were based on MIC results. Do the authors know the minimal bactericidal concentration for this defensin? Moreover, considering that all experiments are time dependent, a time-kill kinetics assay is highly encouraged.
Also in this topic, the discussion has some ups and downs. From lines 267 to 279 there is too much speculation, without a clear comparison with previous literature on defensins. For instance, the authors suggest disulfide bonds reduction in the due to the cytoplasm constituents… It should be at least evaluated in vitro with a simple experiment.
Lots is mentioned about the peptide’s interaction with OM and CM. LUVs containing cardiolipin and LPS mixed with POPE and POPG mimic IM and outer membrane OM in bacteria, respectively. To complement and support the authors data, SPR and leakage assays using different LUVs are strongly encouraged. It will help the authors discuss the preference of this particular defensin for different membrane-like environments. Moreover, since the 3D structure for this defensin is already available, molecular dynamics simulations using bilayer systems could support some of the data here reported.
Most figures show only sytox and GFP images. What about the ones with propidium iodide? This would give us a better idea which cells are viable or not. In figure 4 the authors discuss about cell viability only be means of “explosion”. However, we cannot discard the fact that some bacterial cells may be inviable even though not showing define morphological damages.
Line 306: “Roughly 5-10% of bacteria have exploded within 1 h at 4 μM of L-AvBD103b”. This is the MIC. What about at the MBC? This should have been evaluated. Mostly, because the idea of translating AMPs to the clinic mainly includes eradicating bacterial cells, not only compromising a few and interrupting the growth of other, which could lead to bacterial resistance during treatment.
From lines 313 – 326. Once again, the authors affirm they can’t conclude much in terms of mechanisms of action, but the number of speculations is high and the study lacks additional experiments to support some of these conclusions.
Table 1 caption: “in the exact same conditions” hard to know that based only on the experimental procedures’ description in the other papers. I suggest the authors to remove it.
One of the advantages of using cysteine-rich AMPs is their higher resistance to proteolysis and, therefore, bioavailability in the host. Although the data in table 1 is interestingly, this table must include more data regarding constrained peptides, including others defensins, cyclotides, and other cysteine-rich AMPs (which resemble more to the peptide dscribed in the present study). After all, from the beginning the authors claimed that linear peptides work completely different from defensins… So, the table should be updated accordingly, as for the discussion.
Please update the conclusions after considering the comments above.
Finally, although it is more of a proof-of-concept and methodological study, the authors must include in their discussion the importance of defensins as possible peptide-based drugs for treating bacterial infections. What are the benefits of this methodology if defensins are highly constrained peptides, with modest antibacterial properties and with a costly and time-consuming chemical synthesis? This must be addressed in the manuscript.
Round 2
Reviewer 2 Report
The authors have addressed all my suggestions accordingly, improving the manuscript. My final recommendation is to accept it in its present form.